cellular biology/health and disease and epidemiology

osteoclast, bone resorption, piceatannol, apoptosis, RANKL

**Authors for correspondence:**
Dattatrya Shetti
e-mail: dattakapilshetti@gmail.com
Kun Wei
e-mail: weikun@scut.edu.cn

# Piceatannol attenuates RANKL-induced osteoclast differentiation and bone resorption by suppressing MAPK, NF-κB and AKT signalling pathways and promotes Caspase3-mediated apoptosis of mature osteoclasts

Liuliu Yan, Lulu Lu, Fangbin Hu, Dattatrya Shetti and Kun Wei

School of Biology and Biological Engineering, South China University of Technology, Guangzhou, Guangdong 510006, People's Republic of China

LY, 0000-0002-4640-1016; DS, 0000-0001-8638-804X

Osteoclasts are multinuclear giant cells that have unique ability to degrade bone. The search for new medicines that modulate the formation and function of osteoclasts is a potential approach for treating osteoclast-related bone diseases. Piceatannol (PIC) is a natural organic polyphenolic stilbene compound found in diverse plants with a strong antioxidant and anti-inflammatory effect. However, the effect of PIC on bone health has not been scrutinized systematically. In this study, we used RAW264.7, an osteoclast lineage of cells of murine macrophages, to investigate the effects and the underlying mechanisms of PIC on osteoclasts. Here, we demonstrated that PIC treatment ranging from 0 to 40 μM strongly inhibited osteoclast formation and bone resorption in a dose-dependent manner. Furthermore, the inhibitory effect of PIC was accompanied by the decrease of osteoclast-specific genes. At the molecular level, PIC suppressed the phosphorylation of c-Jun N-terminal kinase (JNK), extracellular signal-regulated kinase (ERK1/2), NF-κB p65,

IκBα and AKT. Besides, PIC promoted the apoptosis of mature osteoclasts by inducing caspase-3 expression. In conclusion, our results suggested that PIC inhibited RANKL-induced osteoclastogenesis and bone resorption by suppressing MAPK, NF-κB and AKT signalling pathways and promoted caspase3-mediated apoptosis of mature osteoclasts, which might contribute to the treatment of bone diseases characterized by excessive bone resorption.

# 1. Introduction

Bone homeostasis is maintained by the precise balance between bone formation and bone resorption, which are carried out by osteoblasts and osteoclasts, respectively [1]. However, enhanced osteoclastic bone resorption can cause imbalance of bone homeostasis, which leads to many bone-destructive diseases including osteoporosis, rheumatoid arthritis and metastatic cancers [2–4]. Osteoclasts originate from the haematopoietic progenitor cells of monocyte/macrophage lineage which undergo differentiation to form specialized bone resorbing cells [5,6]. The process is essentially determined by two key cytokines, macrophage colony stimulation factor (M-CSF) and receptor activator of NF-κB ligand (RANKL), respectively [7,8]. Notably, mouse monocyte/macrophage cell lineage (RAW264.7) can be induced to differentiate into osteoclasts by RANKL in the absence of M-CSF [9,10].

The binding of RANKL to its receptor RANK leads to the recruitment of tumour necrosis factor receptor-associated factor 6 (TRAF6), which then activates several downstream signalling pathways including NF-κB, MAPKs (ERK, JNK and p38) and Src/PI3K/AKT [11]. These events ultimately lead to the expression of NFATc1, a master regulator for osteoclastogenesis [12]. NFATc1 plays an important role in osteoclast formation and function by regulating a number of osteoclast-specific genes, including tartrate-resistant acid phosphatase (TRAP), cathepsin K (CTSK), matrix metalloproteinase 9 (MMP-9) and dendritic cell-specific transmembrane protein (DC-STAMP) [11]. Thus, the agents that suppress RANKL-mediated signalling pathways can be a potential therapeutic target.

Piceatannol (PIC) is a polyphenolic stilbene found naturally in various plants, including grapes, rhubarb and sugarcane [13,14]. Pharmacological effects of PIC such as antioxidative, anti-inflammatory, anti-cancer and cardioprotective properties have been reported before [15]. Specifically, studies have confirmed that PIC exhibits the representative anti-inflammatory effect through the NF-κB, MAPKs and PI3K/AKT pathways [16–18], which are also essential for osteoclast differentiation and bone resorption. Based on existing and confirmed knowledge, only a few studies have reported the direct effect of PIC on osteoclast formation. Jia et al. [19] revealed that PIC inhibited the activation of Syk and the osteoclast formation. Ke et al. [20] demonstrated that PIC decreased the expression of miR-183, resulting in suppressed osteoclastogenesis. Although these findings have provided an understanding of the effects of PIC on osteoclast formation, the precise underlying mechanism of PIC on osteoclast differentiation and bone resorption from RANKL-induced RAW264.7 cells remains unclear. Moreover, the effects and the molecular mechanism of PIC on the survival of mature osteoclasts have not been investigated. In this study, we investigate the direct effects and the underlying mechanism of PIC on the differentiation, function and survival of osteoclasts.

# 2. Material and methods

## 2.1. Reagents and antibodies

Dulbecco's modified Eagle's medium (DMEM) and fetal bovine serum (FBS) were purchased from GIBCO (Invitrogen Corp., Carlsbad, CA, USA). The M-CSF and RANKL were obtained from R & D Systems (Minneapolis, MN, USA). The cell counting kit-8 was obtained from Phygene (Fuzhou, China). The caspase-3 activity assay kit and Hoechst 33258 were obtained from Beyotime Biotechnology (Shanghai, China). TRAP stain kit was purchased from Sigma-Aldrich (USA). RAW264.7 cells were obtained from Procell (Wuhan, China). Piceatannol (PIC) was acquired from Selleck Chemicals (USA) and dissolved in dimethyl sulfoxide (DMSO) and then diluted to the required working concentrations in complete culture medium. All antibodies used in this study were obtained from Cell Signalling Technology (Danvers, MA).

## 2.2. Cell culture and treatment

RAW264.7 cells were grown in a dish with complete culture medium composed of 90% DMEM, 10% FBS and 1% penicillin/streptomycin. The cells were put in an incubator with a humidified atmosphere containing 5% $CO_2$ at 37°C. Cells were treated with RANKL (20 ng ml$^{-1}$), M-CSF (10 ng ml$^{-1}$) and different concentrations of PIC (0, 2.5, 5, 10, 20 and 40 µM) to induce osteoclast formation. The highest concentration of DMSO was below 0.1% during the experiments, and this concentration served as vehicle control.

## 2.3. Cytotoxicity assay

Cell viability was detected with the CCK-8. Briefly, RAW264.7 cells were seeded into sterile 96-well plates at a density of $5 \times 10^3$ cells well$^{-1}$. After 24 h, cells were treated with different concentrations of PIC (2.5, 5, 10, 20, 40 µM) or vehicle (0.1% DMSO) for 48 or 72 h. At the end of the culture period, the culture media were replaced with 100 µl mixture of 10% CCK-8 and 90% DMEM. After cells were incubated for 2 h, the absorbance values were measured using a 96-well plate reader at 450 nm.

## 2.4. TRAP staining and TRAP activity assay

RAW264.7 cells were seeded into sterile 96-well plates at $5 \times 10^3$ cells per well in complete medium for 24 h until they attached to the plates. Cells were then cultured with the complete culture medium supplemented with 20 ng ml$^{-1}$ RANKL, 10 ng ml$^{-1}$ M-CSF and various non-cytotoxic concentrations of PIC (2.5, 5, 10, 20, 40 µM) or vehicle (0.1% DMSO) for 4 days. At the end of the culture period, the cells were fixed with 4% paraformaldehyde for 20 min and then stained for TRAP according to the manufacturer's instructions. TRAP-positive cells containing three or more nuclei were counted as osteoclasts. Images were obtained using an Olympus IX83 inverted microscope. TRAP activity was performed as previously described [21] and was quantified by detecting optical absorbance at 405 nm using EnSpire 2300 multimode reader (PerkinElmer).

## 2.5. Bone resorption assay

To evaluate bone resorption, RAW264.7 cells were seeded on bone slices in 24-well plates at a density of $1.5 \times 10^4$ cells per well, in the presence of RANKL (20 ng ml$^{-1}$), M-CSF (10 ng ml$^{-1}$) and varying concentrations of PIC (10, 20, 40 µM) or vehicle (0.1% DMSO). Afterwards, the attached cells were cultured for 5 days (SEM) and 7 days (toluidine blue). The media were replaced every 2 days. At the end of the culture period, cells that had attached to bone slices were washed by mechanical agitation. Bone slices images were taken by using a scanning electron microscope (SEM, FEI Q25) at 10 kV. However, toluidine blue staining was also used for bone resorption pit assay as previously described [22]. Resorbed areas on bone slices were analysed with IMAGEJ software.

## 2.6. Real-time qPCR

RAW264.7 cells were seeded at a density of $4 \times 10^4$ cells per well in a 12-well plate and were exposed to PIC (20, 40 µM) or vehicle (0.1% DMSO) in the presence of RANKL (20 ng ml$^{-1}$) and M-CSF (10 ng ml$^{-1}$) for 4 days. Total RNA was extracted from cultured cells using UNIQ-10 Column Trizol Total RNA Isolation Kit (Sangon Biotech). Single-stranded cDNA was synthesized from 1 µg of total RNA using reverse transcriptase (Toyobo, Japan). Real-time qPCR was performed with ChamQ SYBR qPCR Master Mix (Q331-02, Vazyme) using ABI 7500 Real-time quantitative PCR machine (Applied Biosystems, Foster City, CA). All reactions were run in triplicate. The primers (Sangon Biotech) used in this study are as follows: GAPDH, 5′-GGTTGTCTCCTGCGACTTCA-3′ and 5′-TGGTCCAGGGTTTCT TACTCC-3′; MMP-9, 5′-CAAAGACCTGAAAACCTCCAAC-3′ and 5′-GACTGCTTCTCTCCCATCATC-3′; CTSK, 5′-GGCCAGTGTGGTTCCTGTTGG-3′ and 5′-CCGCCTCCACAGCCATAATTCTC-3′; TRAP, 5′-CAAGAACTTGCGACCATTGTTA-3′ and 5′-ATCCATAGTGAAACCGCAAGTA-3′; DCSTAMP, 5′-CG TTGCCCTGCTCTCTTCTG-3′ and 5′-CAGCCGCAATCAAAGCGTTC-3′; NFATc1, 5′-TACCAGGTC CACCGGATCAC-3′ and 5′-CCCGATGTCTGTCTCCCCTT-3′.

## 2.7. Western blotting

RAW264.7 cells were seeded in a sterile six-well plate at a density of $1 \times 10^6$ cells per well. After pretreatment with PIC (40 µM) or vehicle (0.1% DMSO) for 6 h, the RAW264.7 cells were treated with RANKL (50 ng ml$^{-1}$) for 0, 15 and 30 min. The cells were then washed in ice-cold PBS and lysed in radio-immunoprecipitation assay buffer accompanied with phosphatase and protease inhibitors. The cell lysates were centrifuged at 12 000 r.p.m. for 10 min at 4°C and supernatants were collected. The protein concentrations were measured by a bicinchoninic acid protein assay kit. A total of 30 µg of each protein sample was separated by sodium dodecyl sulfate-polyacrylamide gel electrophoresis and transferred to polyvinylidene difluoride membranes (Millipore, Bedford, MA, USA). Afterwards, the membranes were blocked with 5% skim milk for 2 h, and then incubated with primary antibodies overnight at 4°C, followed by incubation with the appropriate secondary antibodies. Antibody reactivity was detected using the ProteinSimple FluorChem M Imaging System as recommended by the manufacturer.

## 2.8. Mature osteoclasts

RAW264.7 cells were cultured in DMEM complete medium supplemented with RANKL (20 ng ml$^{-1}$) and M-CSF (10 ng ml$^{-1}$) for 4 days to differentiate into mature osteoclasts. The media were changed on day 2. On day 4, mature osteoclasts were then cultured with PIC or vehicle (0.1% DMSO) for 24 h.

## 2.9. Mature osteoclast survival assay

Mature osteoclasts were cultured with PIC (20, 40 µM) or vehicle (0.1% DMSO) for 24 h. At the end of treatment, the cells were fixed with 4% paraformaldehyde for 20 min, followed by TRAP staining. TRAP-positive multinucleated cells containing three or more nuclei were counted as mature osteoclasts.

## 2.10. LDH assay

Necrosis is a type of cell death characterized by the breakdown of cell plasma membranes, which eventually causes the release of intracellular contents such as lactate dehydrogenase (LDH) into extracellular milieu [23]. After the culture period, LDH release was performed as previously described [23].

## 2.11. Hoechst staining

Mature osteoclasts were cultured with PIC (40 µM) or vehicle (0.1% DMSO) for 24 h. At the end of culture period, the cells were washed with PBS gently and stained with Hoechst 33258 in the dark for 15 min. Photomicrographs were captured using a Zeiss Ism710 Confocal Laser Scanning Microscope.

## 2.12. Caspase-3 activity assay

Mature osteoclasts were exposed to PIC (40 µM) or vehicle (0.1% DMSO) for 24 h. After the culture period, an assay kit was used to measure caspase-3 activity of the cells as recommended by the manufacture. Caspase-3 activity was determined by measuring optical absorbance at 405 nm using EnSpire 2300 Multimode Reader (PerkinElmer).

## 2.13. Statistical analysis

All experimental data expressed as the mean ± s.d. Statistical significance was determined using Student's paired $t$-test, one-way analysis of variance and GRAPHPAD PRISM 6.01 software. In all cases, $p < 0.05$ was considered significant.

# 3. Results

## 3.1. PIC inhibits RANKL-induced osteoclast formation

To determine the effect of PIC on osteoclast differentiation, RAW264.7 cells were treated with different concentrations of PIC. We first examine the cytotoxicity effect of PIC on RAW264.7 cells. The various

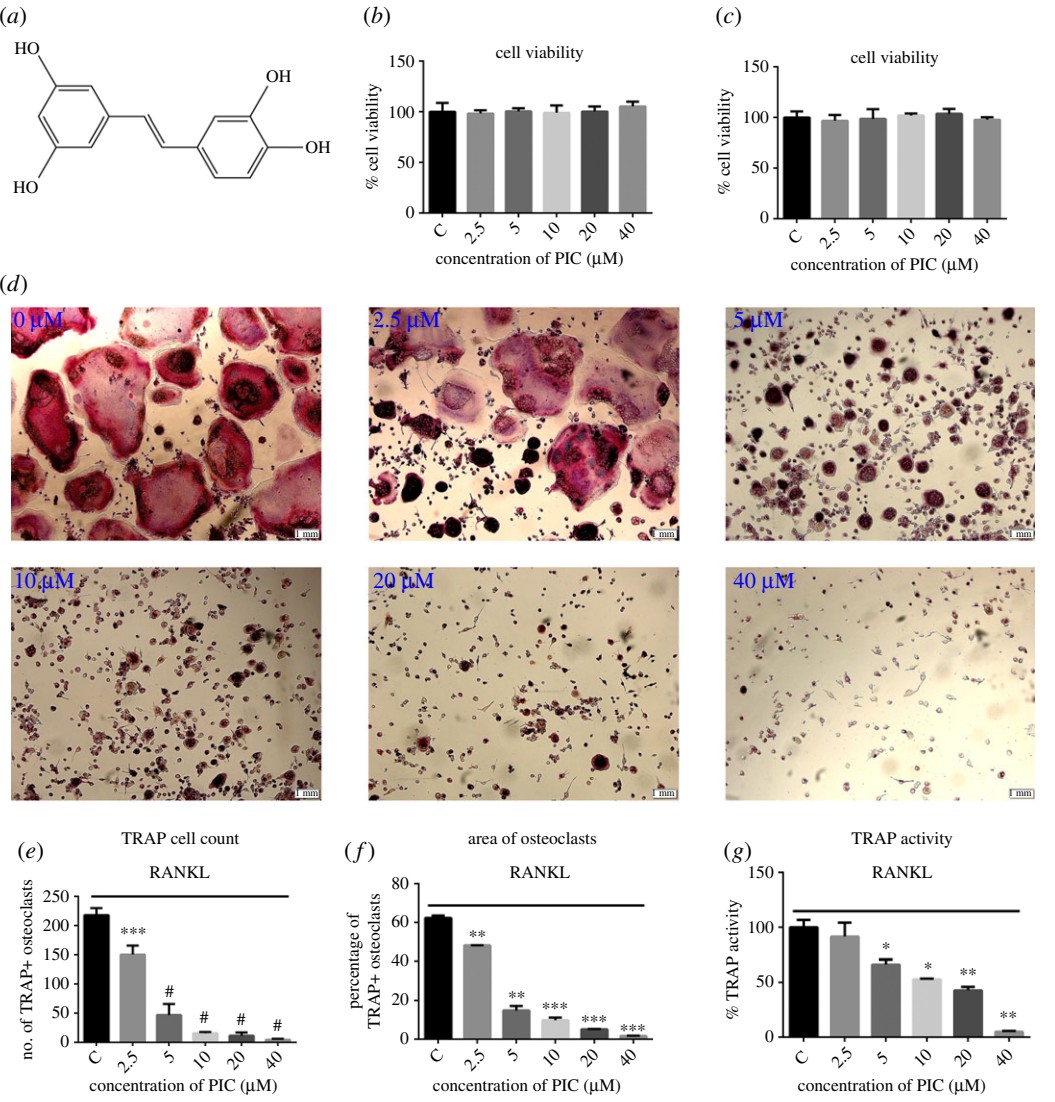

**Figure 1.** PIC attenuates RANKL-induced osteoclast differentiation. (*a*) The chemical structure of PIC; RAW264.7 cells were treated with various concentrations of PIC (2.5, 5, 10, 20, 40 μM) or vehicle (0.1% DMSO) at the different time interval for 48 h (*b*) and 72 h (*c*), and CCK-8 was used to measure cell viability; (*d*) RAW264.7 cells were differentiated into osteoclasts in the presence of M-CSF (10 ng ml$^{-1}$), RANKL (20 ng ml$^{-1}$) and PIC (0, 2.5, 5, 10, 20, 40 μM) for 4 days. Then cells were fixed with 4% paraformaldehyde for 20 min and stained by TRAP; (*e*) TRAP-positive cells in a well; (*f*) the area of TRAP-positive multinucleated cells; (*g*) TRAP activity quantification (*$p < 0.05$, **$p < 0.01$, ***$p < 0.001$, #$p < 0.0001$).

concentrations used in our studies showed no significant effect on cell viability (figure 1*b,c*). RAW264.7 cells were then differentiated with M-CSF and RANKL for 4 days in the presence of various concentrations of PIC. The number of TRAP-positive osteoclasts increased in vehicle control cells and significantly decreased after PIC treatment in a dose-dependent manner (figure 1*d*−*f*). Furthermore, PIC completely inhibited osteoclast formation at a concentration of 40 μM and decreased the TRAP activity in a dose-dependent manner (figure 1*g*).

## 3.2. PIC attenuates osteoclastic bone resorption

Even though PIC could inhibit osteoclast formation, it was unclear whether PIC has the similar effect on bone resorption. Thus, we first performed pit formation assay on bovine slices by using SEM. In control, several bone resorption pits were observed (figure 2*a*). The resorption area was significantly decreased with an increase in PIC concentration (figure 2*b*). We next performed the pit formation assay using toluidine blue staining on bovine slices for more convincing results. As shown in figure 2*c*, an

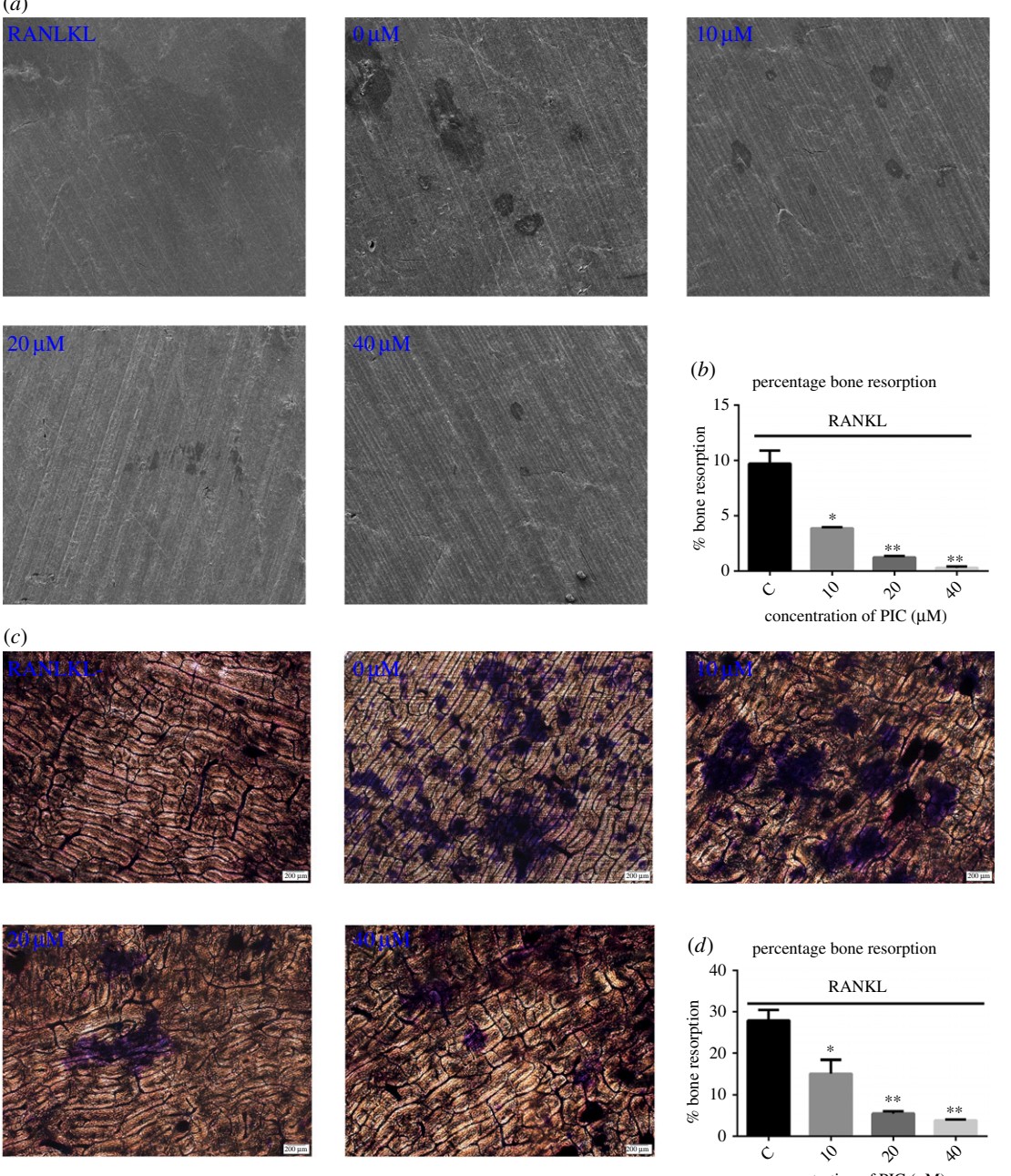

**Figure 2.** PIC inhibits osteoclast-mediated bone resorption. (*a*) RAW264.7 cells were grown on bone slices in a 24-well plate and cultured with DMEM complete medium containing RANKL (20 ng ml$^{-1}$), M-CSF (10 ng ml$^{-1}$) and various concentrations of PIC (0, 10, 20, 40 μM) for 5 days. Bone resorption images were taken by SEM; (*b*) the resorbed areas shown in (*a*) were quantified using ImageJ software; (*c*) RAW264.7 cells were grown on bone slices in a 24-well plate and cultured with DMEM complete medium containing RANKL (20 ng ml$^{-1}$), M-CSF (10 ng ml$^{-1}$) and various concentrations of PIC (0, 10, 20, 40 μM) for 7 days. Representative images were taken after toluidine blue staining; (*d*) the resorbed areas shown in (*c*) were quantified using ImageJ software (*$p < 0.05$, **$p < 0.01$).

obvious increase of bone resorption pits was seen in the control group. By contrast, bone resorption pits significantly decreased with PIC in a dose-dependent manner (figure 2*d*).

## 3.3. PIC inhibits RANKL-stimulated osteoclast-specific gene expression

To further identify the inhibitory effect of PIC on RANKL-induced osteoclast differentiation and bone resorption, we performed real-time qPCR to examine the expression of RANKL-induced osteoclast-

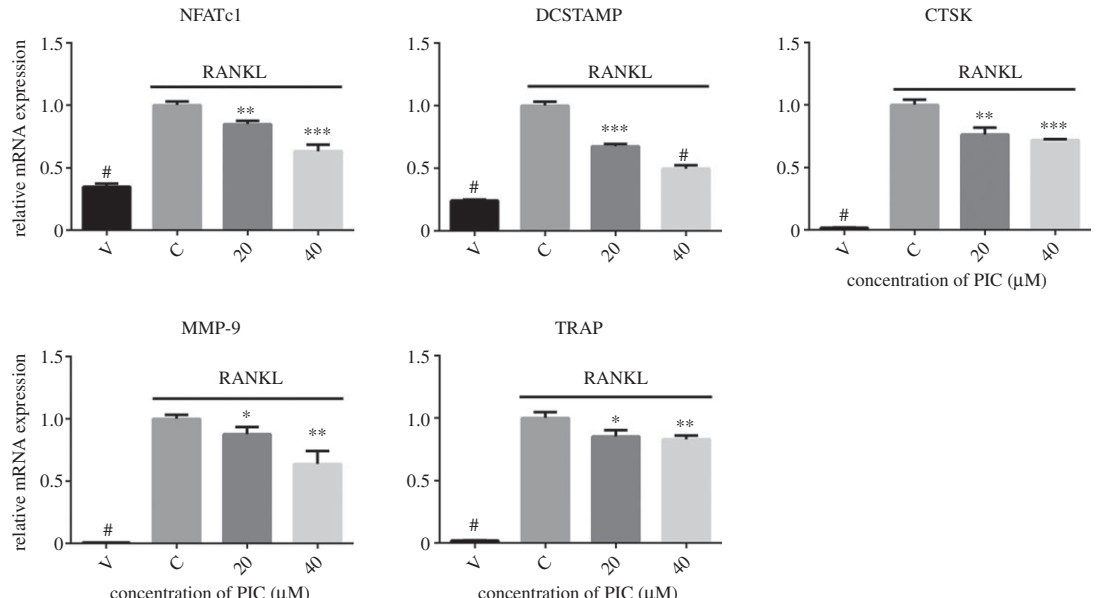

**Figure 3.** PIC attenuates osteoclast-specific gene expression. RAW264.7 cells were cultured with PIC (20, 40 μM) or vehicle (0.1% DMSO) in combination with RANKL (20 ng ml$^{-1}$) and M-CSF (10 ng ml$^{-1}$) for 4 days. Total RNA was isolated with Trizol and reverse transcribed into cDNA. The relative mRNA expression of NFATc1, DCSTAMP, CTSK, MMP-9 and TRAP was analysed by real-time qPCR. Results are shown relative to the RANKL-treated control (*$p < 0.05$, **$p < 0.01$, ***$p < 0.001$, $^{\#}p < 0.0001$).

specific genes. As expected, RANKL significantly induced the expression of NFATc1, DCSTAMP, CTSK, MMP-9 and TRAP. However, the mRNA expression of these genes was effectively reduced by PIC in a concentration-dependent manner (figure 3).

## 3.4. PIC suppresses RANKL-stimulated activation of NF-κB, JNK, ERK and AKT

To elucidate the intracellular mechanism of the inhibitory effect of PIC on osteoclast formation and bone resorption, we investigated the RANKL-mediated MAPKs, NF-κB and AKT signalling pathways. RAW264.7 cells were pretreated with PIC or vehicle control for 6 h. Cells were then treated with RANKL (50 ng ml$^{-1}$) for different time (0, 15, 30 min). MAPKs, NF-κB and AKT signalling pathways were evaluated by immunoblotting. The results showed that the expression levels of JNK, ERK, p38 and AKT were nearly similar among control and PIC groups. Furthermore, the phosphorylated proteins (p-JNK, p-ERK, p-AKT, p-p65) were marginally expressed without RANKL treatment (0 min), while these proteins were highly expressed in RANKL-stimulated control group (15 and 30 min). Notably, the phosphorylated protein (p-IκBα) maintained high levels of expression in RANKL-treated control group (0, 15 and 30 min). However, the phosphorylation of JNK, ERK, AKT, IκBα and p65 were greatly reduced by PIC pretreatment (figure 4a,b). Further studies showed that p38 and p-p38 were not affected by PIC pretreatment (figure 4c).

## 3.5. PIC promotes the apoptosis of mature osteoclasts

To evaluate the influence of PIC on osteoclast apoptosis, mature osteoclasts were exposed to PIC for 24 h and stained for TRAP. We found that PIC treatment attenuated the survival of mature osteoclasts in a dose-dependent manner (figure 5a,b). To investigate whether the decrease in mature osteoclast survival was accompanied by apoptosis, LDH release for cell necrosis and Hoechst 33258 staining for nuclear fragmentation were performed as described in the methods. As shown in figure 5c, mature osteoclasts did not release significant LDH after 24 h exposure to PIC. On the other hand, an increasing nuclear fragmentation was observed in the PIC-treated cells compared to the control, indicating that PIC treatment enhanced apoptosis of mature osteoclasts (figure 5d). Consistent with its pro-apoptotic effect, addition of PIC increased caspase-3 activity and induced the cleavage of the caspase-3 precursor (figure 5e,f).

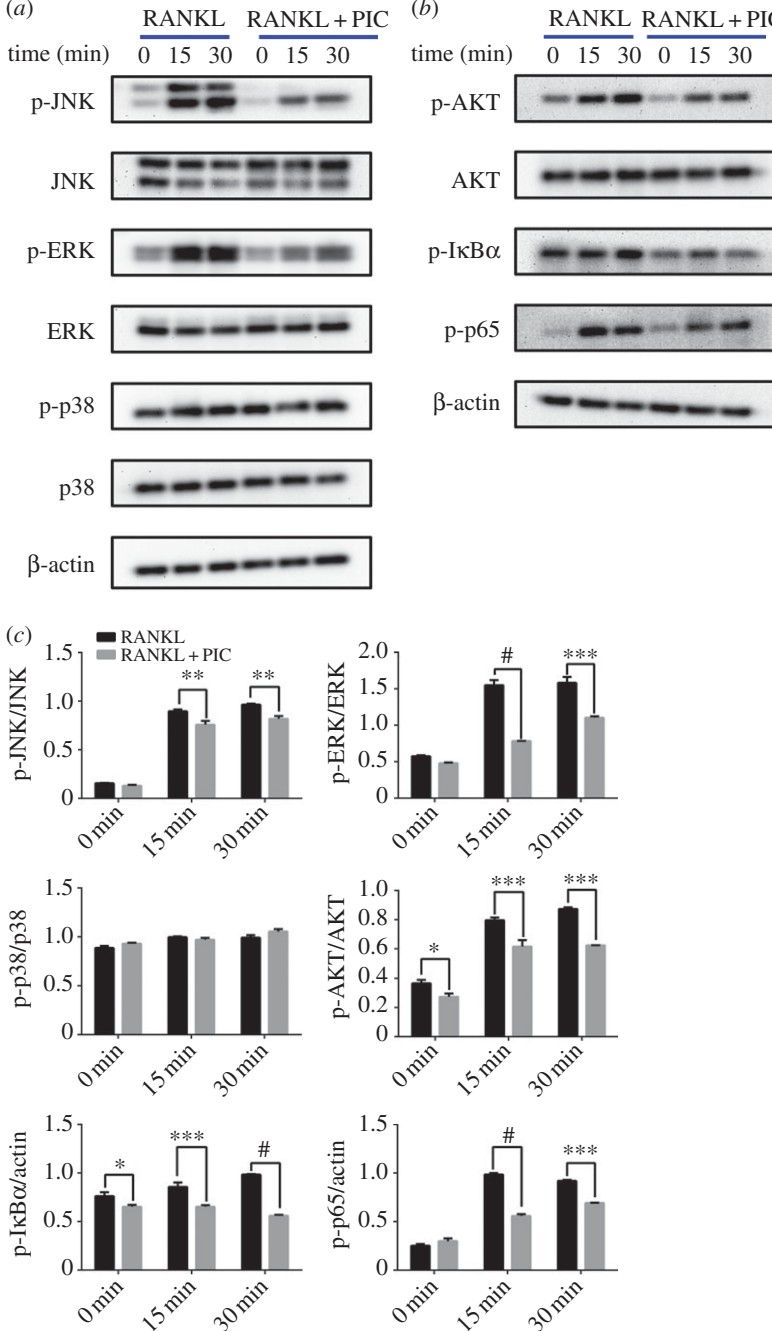

**Figure 4.** PIC attenuates RANKL-stimulated NF-κB, JNK, ERK and AKT signalling pathways. (a,b) RAW264.7 cells were pretreated with PIC (40 μM) or vehicle control for 6 h and then treated with RANKL (50 ng ml$^{-1}$) for different time. Immunoblotting was conducted to determine the protein levels. Antibody against β-actin served as the loading control. (c) Quantitative analysis of phosphorylation of JNK, ERK, p38, AKT, IκBα and p65 were determined by IMAGEJ. Results are shown compared with the RANKL-treated control group (*$p < 0.05$, **$p < 0.01$, ***$p < 0.001$, #$p < 0.0001$).

## 4. Discussion

Osteoclasts are unique bone resorptive cells which are essential for skeletal metabolism. Increased bone resorption can cause the disorder of bone homeostasis, such as osteoporosis. Therefore, inhibition of the formation and function of osteoclasts may provide a promising therapeutic method for osteoclast-based diseases. In the present study, we found that PIC could significantly inhibit RANKL-induced osteoclast formation and bone resorption. Furthermore, PIC inhibited the activation of NF-κB, JNK, ERK and AKT. Notably, for the first time, we found that PIC promoted the apoptosis of mature osteoclasts via caspase-3

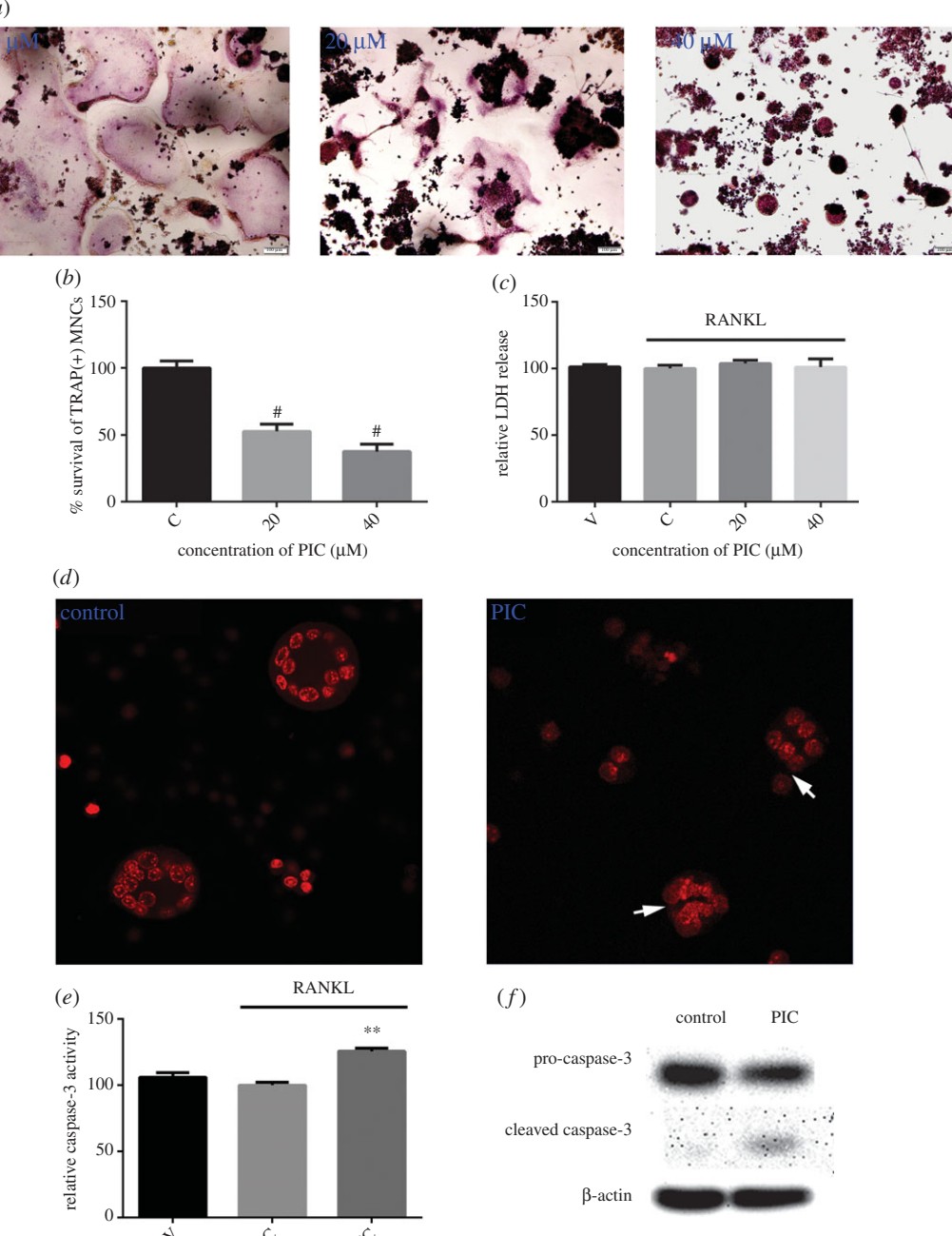

**Figure 5.** PIC promotes caspase3-mediated apoptosis of mature osteoclasts. Mature osteoclasts were treated with PIC or vehicle for 24 h. (a) Cells were fixed with 4% paraformaldehyde and stained for TRAP; (b) the survival of TRAP-positive MNCs relative to the control group; (c) LDH release for necrosis; (d) Hoechst 33258 staining was performed to visualize nuclear fragmentation. Apoptotic mature osteoclasts are indicated with white arrows; (e) caspase-3 activity assay for apoptosis was performed; (f) caspase-3 and cleaved caspase-3 protein levels were determined by Western blotting (**$p < 0.01$, #$p < 0.0001$).

activation. Taken together, we concluded that PIC might be valuable as a potent therapeutic agent for treating osteoclast-based metabolic bone diseases.

Many plants are rich in polyphenols, which may be beneficial to bone health. Representative studies have indicated a positive relationship between intake of polyphenols and bone health [24–27]. Previous studies showed that ferulic acid, a dietary polyphenol, attenuated RANKL-induced osteoclast formation and bone resorption through the inhibition of NF-κB signalling pathway [28]. Furthermore, Shahnazari et al. found that dietary dried plum increased bone volume and strength, and indicated that these effects might be linked to the immune system and dried plum-specific polyphenols [29]. Piceatannol, a naturally occurring polyphenolic stilbene found in various plants, has been confirmed to exhibit anti-inflammatory

and anti-cancer properties [30]. To our knowledge, there were almost no studies reporting the effect and the underlying mechanism of PIC on osteoclast-associated diseases. In this study, to explore the direct effects and its underlying mechanism of PIC on the formation, function and survival of osteoclasts, the relevant signalling pathways of osteoclast differentiation and the apoptosis of mature osteoclasts were investigated.

Signalling mediated by MAPKs, including JNK, ERK and p38, plays a key role in osteoclast differentiation and function [31]. To date, many polyphenols derived from natural plants have been demonstrated to inhibit osteoclast formation and function by suppressing the MAPK signalling pathway. For example, eriodictyol, a natural occurring flavonoid found in citrus fruits, inhibited osteoclast formation by suppressing MAPK-NFATc1 signalling pathways [32]. Kaempferol, the most common flavonoid present in a variety of plants, impaired IL-1β-stimulated, RANKL-mediated osteoclast differentiation via the MAPK signalling pathway [33]. Consistently, in this study, we also confirmed that PIC exerts antiosteoclastogenesis and antiresorptive effects via suppression of the phosphorylation of JNK and ERK in RANKL-stimulated RAW264.7 cells.

The binding of RANKL to its receptor RANK leads to the activation of PI3K/AKT signalling pathway, which plays an important role in regulating osteoclast survival [34]. In the current study, we found that PIC significantly suppressed the RANKL-stimulated phosphorylation of AKT in RAW264.7 cells. These data imply that PIC displays an inhibitory effect on osteoclast differentiation, which may be due to the decreased survival of osteoclast precursor cells during differentiation because of AKT suppression. The activation of NF-κB signalling pathway is critical for RANKL-induced osteoclast formation and bone resorption, since the deletion of both NF-κB p50 and p52 subunits caused severe osteoporosis due to failure of osteoclast formation [35,36]. Previous studies showed that piceatannol suppressed TNF-induced NF-κB activation in human myeloid cells by inhibiting the phosphorylation of IκBα and p65 [37]. Similarly, in this study, we also observed that PIC attenuated RANKL-induced NF-κB activation in RAW264.7 cells through suppression of IκBα kinase and p65 phosphorylation.

Osteoclast apoptosis was critical for regulating bone homeostasis [34]. Recently, more and more researchers considered the induction of osteoclast apoptosis as a potential therapeutic target for bone diseases characterized by excessive bone resorption [38–40]. In this study, we found that PIC effectively reduced the number of mature osteoclasts, which suggested that PIC could suppress the survival of mature osteoclasts. Nuclear fragmentation is a key marker of apoptosis [41]. To investigate whether the decline in the survival rate of mature osteoclasts was accompanied by apoptosis, Hoechst 33258 staining was performed to observe the nuclear fragmentation in mature osteoclasts, and the LDH release was performed to exclude the necrosis. Mature osteoclasts did not release significant LDH, which suggested that apoptosis was the leading cause of cell death. Caspase-3 is involved in the majority of apoptotic effects, and is likely to be critical for osteoclast differentiation [42]. Similarly, in this study, we found that PIC increased caspase-3 activity and induced the cleavage of the caspase-3 precursor. These results indicated that the activation of caspase-3 might be associated with PIC-stimulated osteoclast apoptosis. However, the precise mechanism of PIC-induced apoptosis of mature osteoclasts remains to be investigated.

# 5. Conclusion

In summary, our findings show that PIC significantly suppresses the formation and function of osteoclasts. Furthermore, PIC promotes the apoptosis of mature osteoclasts. Therefore, PIC has potential therapeutic effects on osteoclast-based bone diseases.

Data accessibility. Data available from the Dryad Digital Repository: https://doi.org/10.5061/dryad.df19750 [43].
Authors' contributions. L.Y. and F.H. designed the experiments; L.Y. and L.L. performed experiments; L.Y. wrote the rough draft of manuscript; K.W. and D.S. edited the manuscript. All authors gave final approval for publication and agree to be held accountable for the work performed therein.
Competing interests. The authors declare no competing interests.
Funding. This research was funded by Multi-stage construction of high-strength microsphere scaffolds and development of bone repair materials (grant no. 2015A020214005) and the International Cooperation Projects of Guangdong Provincial Science and Technology (grant no. 2015A050502013).
Acknowledgements. The authors thank Y.G.Y., F.H.K. and C.L.M. for their technical support.

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
