## [Reviewer comments · Royal Society Open Science]

Review History

RSOS-190360.R0 (Original submission)

Review form: Reviewer 1

Is the manuscript scientifically sound in its present form?

Yes

Are the interpretations and conclusions justified by the results?

Yes

Is the language acceptable?

Yes

Is it clear how to access all supporting data?

Yes

Do you have any ethical concerns with this paper?

No

Have you any concerns about statistical analyses in this paper?

No

Recommendation?

Accept with minor revision (please list in comments)

Comments to the Author(s)

This study investigated that PIC inhibited RANKL-induced osteoclastogenesis and bone resorption by suppressing MAPK, NF- κ B and AKT signaling pathways and promoted caspase3-mediated apoptosis of mature osteoclasts. The previous study declared that PIC inhibits the NF- κ B, MAPK and PI3K pathways. PIC also inhibits the formation of osteoclasts (doi: 10.1096/fj; doi: 10.1016/j), so the innovation of this research is relatively insufficient. At the same time, some problems should be addressed in this study.

Major concerns:

1. The osteoclastogenesis assay not only implemented by RAW264.7 cells, but also should use BMMCs.
2. The toxicity of PIC on RAW264.7 cells should be declared. In Figure 5A, the TRAP staining showed that number of osteoclasts were decreased by PIC, it also may be caused by toxic effect of PIC.
3. The Figure 5 showed that PIC promoted apoptosis of osteoclasts and decreases the number of osteoclasts. Please explain that why the number of osteoclasts not change in cell viability assay (Figure 1B & C).

Minor concerns:

1. The negative control group should be added in Figure 1D.
2. The TRAF6 trigger NF- κ B, MAPK pathway, and the C-Src trigger PI3K pathway. (line 52-58)

Review form: Reviewer 2

Is the manuscript scientifically sound in its present form?

Yes

Are the interpretations and conclusions justified by the results?

Yes

Is the language acceptable?

Yes

Is it clear how to access all supporting data?

Not Applicable

Do you have any ethical concerns with this paper?

No

Have you any concerns about statistical analyses in this paper?

No

Recommendation?

Accept with minor revision (please list in comments)

Comments to the Author(s)

The submitted manuscript claims to treat the bone-destructive diseases by naturally occurring organic polyphenolic stilbene compound called Piceatannol. This compound is present in many foods as a strong antioxidant and anti-inflammatory effect. There was no toxicity effect of Piceatannol on RAW264.7 cells insisting its safe to use them as a drug to inhibit the osteoclast formation and bone resorption. Piceatannol successfully suppressed the MAPK, NF- κ B and AKT signaling pathway which considered to be a major molecular pathway in osteoclast formation and bone resorption. Further, it induces apoptosis in mature osteoclast by caspases 3 dependent pathway. Overall, the research article nearly covers all the significant area which is required to address to treat the bone-destructive diseases.

- 1)I highly recommend this research article to be accepted.
- 2)The research article clearly depicts the importance of the Piceatannol in treating bone-destructive diseases by covering most significant research area
- 3)The research article is novel because it uses the naturally occurring compound to treat bone disease with no cytotoxicity effect
- 4)The minor suggestion is to write a conclusion in the end also discussion part can be written more accurately

Decision letter (RSOS-190360.R0)

07-May-2019

Dear Mr Yan

On behalf of the Editors, I am pleased to inform you that your Manuscript RSOS-190360 entitled "Piceatannol Attenuates RANKL-Induced Osteoclast Differentiation and Bone Resorption and Promotes Caspase3-Mediated Apoptosis of Mature Osteoclasts" has been accepted for publication in Royal Society Open Science subject to minor revision in accordance with the referee suggestions. Please find the referees' comments at the end of this email.

The reviewers and handling editors have recommended publication, but also suggest some minor revisions to your manuscript. Therefore, I invite you to respond to the comments and revise your manuscript.

- Ethics statement

- Data accessibility

It is a condition of publication that all supporting data are made available either as supplementary information or preferably in a suitable permanent repository. The data accessibility section should state where the article's supporting data can be accessed. This section should also include details, where possible of where to access other relevant research materials such as statistical tools, protocols, software etc can be accessed. If the data has been deposited in

an external repository this section should list the database, accession number and link to the DOI for all data from the article that has been made publicly available. Data sets that have been deposited in an external repository and have a DOI should also be appropriately cited in the manuscript and included in the reference list.

<http://datadryad.org/submit?journalID=RSOS&manu=RSOS-190360>

- **Competing interests**

- **Authors' contributions**

- **Acknowledgements**

- **Funding statement**

Because the schedule for publication is very tight, it is a condition of publication that you submit the revised version of your manuscript before 16-May-2019. Please note that the revision deadline will expire at 00.00am on this date. If you do not think you will be able to meet this date please let me know immediately.

on behalf of Dr John Dalton (Associate Editor) and Catrin Pritchard (Subject Editor)
openscience@royalsociety.org

Associate Editor Comments to Author (Dr John Dalton):

Associate Editor: 1

Comments to the Author:

The paper was well received and thought of as novel and well performed. However, one reviewer had a number of relative major and several minor comments that should be addressed before the paper can be accepted.

Reviewer comments to Author:

Reviewer: 1

Comments to the Author(s)

This study investigated that PIC inhibited RANKL-induced osteoclastogenesis and bone resorption by suppressing MAPK, NF- κ B and AKT signaling pathways and promoted caspase3-mediated apoptosis of mature osteoclasts. The previous study declared that PIC inhibits the NF- κ B, MAPK and PI3K pathways. PIC also inhibits the formation of osteoclasts (doi: 10.1096/fj; doi: 10.1016/j), so the innovation of this research is relatively insufficient. At the same time, some problems should be addressed in this study.

Major concerns:

1. The osteoclastogenesis assay not only implemented by RAW264.7 cells, but also should use BMMCs.
2. The toxicity of PIC on RAW264.7 cells should be declared. In Figure 5A, the TRAP staining showed that number of osteoclasts were decreased by PIC, it also may be caused by toxic effect of PIC.
3. The Figure 5 showed that PIC promoted apoptosis of osteoclasts and decreases the number of osteoclasts. Please explain that why the number of osteoclasts not change in cell viability assay (Figure 1B & C).

Minor concerns:

1. The negative control group should be added in Figure 1D.
2. The TRAF6 trigger NF- κ B, MAPK pathway, and the C-Src trigger PI3K pathway. (line 52-58)

Reviewer: 2

Comments to the Author(s)

The submitted manuscript claims to treat the bone-destructive diseases by naturally occurring organic polyphenolic stilbene compound called Piceatannol. This compound is present in many

foods as a strong antioxidant and anti-inflammatory effect. There was no toxicity effect of Piceatannol on RAW264.7 cells insisting its safe to use them as a drug to inhibit the osteoclast formation and bone resorption. Piceatannol successfully suppressed the MAPK, NF- κ B and AKT signaling pathway which considered to be a major molecular pathway in osteoclast formation and bone resorption. Further, it induces apoptosis in mature osteoclast by caspases 3 dependent pathway. Overall, the research article nearly covers all the significant area which is required to address to treat the bone-destructive diseases.

- 1)I highly recommend this research article to be accepted.
- 2)The research article clearly depicts the importance of the Piceatannol in treating bone-destructive diseases by covering most significant research area
- 3)The research article is novel because it uses the naturally occurring compound to treat bone disease with no cytotoxicity effect
- 4)The minor suggestion is to write a conclusion in the end also discussion part can be written more accurately

Author's Response to Decision Letter for (RSOS-190360.R0)

See Appendix A.

Decision letter (RSOS-190360.R1)

13-May-2019

Dear Mr Yan,

I am pleased to inform you that your manuscript entitled "Piceatannol Attenuates RANKL-Induced Osteoclast Differentiation and Bone Resorption and Promotes Caspase3-Mediated Apoptosis of Mature Osteoclasts" is now accepted for publication in Royal Society Open Science.

Kind regards,
Royal Society Open Science Editorial Office

on behalf of Dr John Dalton (Associate Editor) and Catrin Pritchard (Subject Editor)
openscience@royalsociety.org

Appendix A

Dear Editors and Reviewers:

Thank you for your letter and for the reviewers' comments concerning our manuscript entitled "Piceatannol Attenuates RANKL-Induced Osteoclast Differentiation and Bone Resorption by Suppressing MAPK , NF- κ B and AKT Signaling pathways and Promotes Caspase3-Mediated Apoptosis of Mature Osteoclasts". Those comments are all valuable and very helpful for revising and improving our paper. The main corrections in the paper and the responds to the reviewer's comments are as flowing:

Reviewer comments to Author:

Reviewer: 1

Reviewer comment 1: The osteoclastogenesis assay not only implemented by RAW264.7 cells, but also should use BMMCs.

Author reply: Thank you for your great suggestion. We are deeply aware of the necessity of both RAW264.7 cells and BMMCs for osteoclastogenesis research. Due to lack of time, I cannot do the additional experiment on BMMCs but I will consider your precious advice in future for my experiments.

Reviewer comment 2: The toxicity of PIC on RAW264.7 cells should be declared. In Figure 5A, the TRAP staining showed that number of osteoclasts were decreased by PIC, it also may be caused by toxic effect of PIC.

Author reply: Thank you for your advice. It was our negligence that we didn't explore the minimum concentration of PIC which caused toxicity. As we mentioned in section 2.1 and 2.2, PIC was dissolved in DMSO and stored at -20 ° in a concentration of 50mM. Thus, the maximum concentration used in our subsequent experiments couldn't exceed 50 μ M to insure the highest concentration of DMSO was below 0.1% during the experiments. However, we have done Pre-Experiments several times to insure that the maximum concentration we could use (50 μ M) showed no cytotoxic effects in RAW264.7 cells (supplemental Fig1). Considering the above reasons, we finally use 40 μ M as the maximum concentration.

As we mentioned in section 3.5—"We found that PIC treatment attenuated the survival of mature osteoclasts in a dose-dependent manner(Figure 5A, B). **To investigate whether the decrease in mature osteoclast survival was accompanied by apoptosis, LDH release for cell necrosis and Hoechst 33258 staining for nuclear fragmentation were performed** as described in the methods. As shown in Figure 5C, mature osteoclasts didn't release significant LDH after 24h exposure to PIC. On the other hand, an increasing nuclear fragmentation was observed in the PIC treated cells compared to the control, indicating that PIC treatment enhanced apoptosis of mature osteoclasts.(Figure 5D). Consistent with its pro-apoptotic effect, **addition of PIC increased caspase-3 activity and induced the cleavage of the caspase-3 precursor** (Figure 5E, F).", the number of mature osteoclasts were decreased by PIC was due to

apoptosis shown by LDH release assay, Hoechst 33258 staining, Caspase-3 activity assay and Caspase-3 protein expression.

Reviewer comment 3: The Figure 5 showed that PIC promoted apoptosis of osteoclasts and decreases the number of osteoclasts. Please explain that why the number of osteoclasts not change in cell viability assay (Figure 1B & C)

Author reply: Thank you for your good comments. Figure 1B&C only represent the cell viability of RAW264.7 cells rather than the mature osteoclasts. As we mentioned in section 2.8—“RAW264.7 cells were cultured in DMEM complete medium supplemented with RANKL (20ng/ml) and M-CSF (10ng/ml)] for 4 days to differentiate into mature osteoclasts”, our results showed that PIC had no effect on the cell viability of RAW264.7 cells, but when RAW264.7 cells differentiated into mature osteoclasts, it induced apoptosis of mature osteoclasts.

Reviewer comment 4: The negative control group should be added in Figure 1D

Author reply: Thank you for your mention. We added the negative control group in supplementary data (supplemental Fig2).

Reviewer comment 5: The TRAF6 trigger NF- κ B, MAPK pathway, and the C-Src trigger PI3K pathway. (line 52-58)

Author reply: Thank you for your great suggestion. We are very sorry for our incorrect writing “which then triggers the activation of several downstream signaling pathways including NF- κ B, MAPKs(ERK, JNK and p38) and PI3K/AKT”. It is noteworthy that several reviews report that Src/PI3K/AKT pathway is one of the downstream signaling pathways of RANKL/RANK/TRAF6 (DOI: 10.1038/nature01658; DOI: 10.1016/S0006-291X(03)00695-8; DOI: 10.1016/j.intimp.2016.04.024). In view of this, we modified the sentence into “which then activates several downstream signaling pathways including NF- κ B, MAPKs(ERK, JNK and p38) and Src/PI3K/AKT”.

Special thanks to you for your good comments.

Reviewer comments to Author:

Reviewer: 2

Reviewer comment 1: The minor suggestion is to write a conclusion in the end also discussion part can be written more accurately

Author reply: Thank you for your great suggestion. We will write a conclusion in the end.